# The Effect of Glycerol-Based Suspensions on the Characteristics of Resonators Excited by a Longitudinal Electric Field

**DOI:** 10.3390/s23020608

**Published:** 2023-01-05

**Authors:** Alexander Semyonov, Boris Zaitsev, Andrey Teplykh, Irina Borodina

**Affiliations:** Kotel’nikov Institute of Radio Engineering and Electronics of RAS, Saratov Branch, Saratov 410019, Russia

**Keywords:** liquid sensors, piezoelectric resonator with a longitudinal electric field, suspensions, resonant frequency, electrical impedance, admittance

## Abstract

This study examines the effect of suspensions based on pure glycerol and diamond powder with different concentrations on the characteristics of resonators with a longitudinal electric field. We used two disk resonators made of the quartz and langasite plates with round electrodes on both sides of the plate and resonant frequencies of 4.4 and 4.1 MHz, operating in shear and longitudinal acoustic modes, respectively. Each resonator was mounted on the bottom of a 30 mL liquid container. During the experiments, the container was filled with the suspension under study in such a way that the resonator was completely immersed in the suspension, and the frequency dependences of the real and imaginary parts of the electrical impedance of the resonator were measured. As a result, the shear modulus of the elasticity and shear coefficient of the viscosity of the studied suspensions were determined. The material constants of the suspensions were found by fitting the theoretical frequency dependences of the real and imaginary parts of the electrical impedance of the resonator to the experimentally measured ones, which was calculated using Mason’s equivalent circuit. As a result, the dependencies of the density, shear modulus of elasticity, shear viscosity coefficient, and velocity of the shear acoustic wave on the volume concentration of the diamond particles were constructed.

## 1. Introduction

The study of the acoustic properties of various suspensions is very important for many areas of science and technology [1,2,3,4,5,6]. Sometimes, there is a need to determine the mechanical properties of liquids as well as mixtures based on liquid and solid particles with high accuracy [7]. A similar need arises when monitoring the state of various reservoirs, wastewater, and groundwater, as well as in areas of science such as biology, medicine, food, pharmaceutical and in the gas- and oil-producing industry [8,9,10,11]. In this case, the choice of sensors is determined by factors such as the reliability, accuracy, commercial availability, and ease of use [12,13,14,15].

Piezoelectric resonators with a longitudinal electric field operating on the bulk acoustic waves (BAW) are still widely used in determining the acoustic properties of both liquids and solids [2,12,15]. Resonators constructed on the basis of AT-cut quartz (YXl 35°25′ cut) and X-cut langasite have often been mentioned in the scientific literature for many years as simple, reliable, and easily accessible acoustoelectric devices [12,14]. Such resonators represent, as a rule, the plane-parallel plate of the piezoelectric crystal with round electrodes on both sides of the plate. The crystallographic orientation and thickness of the piezoelectric plate are determined by the choice of the type of bulk acoustic wave and the operating frequency range.

To study materials using BAW resonators, a good acoustic contact must be provided between the object under study and the surface of the piezoelectric plate [12,13,14]. This is automatically implemented if the object under study is a liquid. This paper presents the results of an experimental study of the effect of the mixture of glycerol and synthetic diamond particles with sizes of 1–2 µm and different volume concentrations on the characteristics of the resonators based on the AT-cut quartz and X-cut langasite, with the round electrodes being placed on both sides of the piezoelectric plate. First, the frequency dependences of the real and imaginary parts of the electrical impedance of the resonator immersed in the suspension were measured. Then, using Mason’s electromechanical equivalent circuit, the shear modulus of the elasticity and shear coefficient of the viscosity of the studied suspensions were determined. The equivalent circuit included the electromechanical transformer and the primary electrical and secondary mechanical circuits. These material constants of the suspensions were determined by fitting the theoretical frequency dependences of the real and imaginary parts of the electrical impedance of the resonator to the experimentally measured ones, which was calculated using the equivalent circuit. As a result, it has been found that the change in the volume concentration of the diamond particles in the suspension leads to a change in its density, shear modulus of elasticity, shear viscosity coefficient, and velocity of the shear longitudinal acoustic wave.

## 2. Preparation of Suspensions and Experimental Procedure

### 2.1. Preparation of Suspensions

The samples of the suspensions of glycerol and synthetic diamond powder with particle sizes of 1–2 µm at different volume concentrations were prepared under the following laboratory conditions. The required amount of diamond powder was added to 30 mL of glycerol and stirred in a magnetic stirrer for 5 h. For research, six different samples of the suspensions were prepared with the volume concentrations of the particles in the mixture at 0.098%, 0.147%, 0.73%, 1.45%, 1.92%, and 2.86%. The density of the obtained samples was determined by weighing a fixed volume (1 mL) of the suspension on an analytical balance (Pioneer PA-214C, OHAUS Corporation, Parsippany, NJ, USA). The volume of the weighed suspension sample was set with a precision laboratory pipette (Kolor, Lenpipet, Saint-Petersburg, Russian Federation). The values of the theoretically expected density of the samples, calculated from the tabular values of the density of glycerol and diamond with the known volume concentration of the powder, turned out to be in good agreement with the measurement results (Figure 1).

The conductivity of the pure glycerol and all suspension samples was measured using an HI8733 conductometer (Hanna, Woonsocket, RI, USA). The measurement results showed that in all cases, the conductivity does not exceed 0.1 µS/cm.

### 2.2. Experimental Methodology

As already noted, for the study of shear and longitudinal acoustic waves, we chose resonators made of the AT-cut quartz and X-cut langasite, respectively. It is known that these cuts of the materials are characterized by low temperature delay coefficients for the pointed bulk modes. These resonators had plate thicknesses of 370 and 706 µm, respectively, and the electrode diameters were 5.8 mm and 7.5 mm, respectively. Each resonator was mounted in the base of a 30 mL plastic container (Figure 2).

Before the measurement, each suspension sample was stirred on a magnetic stirrer for 1 h. Next, the container with the resonator was filled with the studied suspension in such a way that the resonator was completely covered on both sides with the suspension. As the viscosity of the glycerol strongly depends on its temperature [16], before each measurement, a junction of a chromel–alumel thermocouple was lowered into the container with the glycerol. The second junction was set in a Dewar bottle with melting ice at a temperature of 0 °C. The thermocouple was connected to a voltmeter (GDM-78251A, GW Instek, Taiwan), as shown in Figure 2. The resonator with the container was placed in a thermostat with a temperature of 27.3 ± 0.05 °C. The thermocouple junction was immersed in the suspension for no more than 2 min to record the exact value of the suspension temperature.

Then, to eliminate the influence of the thermocouple junction on the measurement results, it was removed from the container, and the frequency dependences of the real and imaginary parts of the electrical impedance of the resonator immersed in the suspension under study were measured. The time of each measurement did not exceed 2 s. These frequency dependences were measured using an E4990A impedance analyzer (Keysight Technologies, Santa Rosa, CA, USA). It should be noted that the temperature of the room with the experimental setup was maintained at 27 ± 0.5 °C with an atmospheric pressure of 760 mm Hg. It was also found that the time of the complete settling of the diamond particles in the glycerol exceeded 24 h at the indicated temperature, and this was well observed visually.

### 2.3. Influence of the Investigated Suspensions on the Characteristics of the Resonator Made of AT-Quartz

It is known that the piezoelectric resonator is characterized by such parameters as the resonant frequency and quality factor of the series and parallel resonances, as well as the maximum values of the real part of the electrical impedance and admittance [9]. Therefore, based on the measurement results, we plotted the dependences of the resonant frequency of the parallel (*F*_par_) and series (*F*_ser_) resonances, the maximum values of the real parts of the electrical impedance (*R*_max_) and admittance (*G*_max_), as well as the quality factors of the parallel (*Q*_par_) and series (*Q*_ser_) resonances on the volume concentration of the powder particles in the suspension for the quartz resonator. These graphs are shown in Figure 3.

The experimental graphs in Figure 3 for the quartz crystal show that in the case of the shear acoustic wave, with the increase in the volume concentration of the diamond particles from 0 to 0.147%, all presented values sharply increase and further monotonically decrease. The exception is the maximum value of the real part of the electrical impedance (Figure 3c), which is characterized by the increase to the volume concentration of 1.45% and the further slight decrease. It should be noted that the small jump corresponding to the concentration of 1.45% observed in almost all graphs in Figure 3 turned out to be below the measurement error, and thus it was not discussed in this paper.

### 2.4. The Influence of the Studied Suspensions on the Characteristics of the Langasite Resonator

The experimental study for the longitudinal wave in the langasite resonator has shown that the maximum values of the electrical impedance and admittance first sharply increase until the volume concentration of particles is 0.098%, then decrease; upon reaching the concentration of 1.45%, they increase again (Figure 4). In this case, the resonant frequencies of the parallel (4.057 MHz) and series (4.06 MHz) resonances did not change with the increase in the volume concentration of the particles in the suspension within the measurement error. A similar situation was observed for the quality factors of the parallel (~16) and series (~16) resonances. This is due to the fact that, unlike a quartz resonator, a langasite resonator resonates on a longitudinal acoustic wave. Therefore, the contact of the resonator with the suspension leads to the emission of a longitudinal acoustic wave into the suspension due to the presence of a mechanical displacement component normal to the surface. Thus, the radiation loss during the contact of the resonator with the suspension significantly reduces the quality factor up to ~16, which does not allow one to accurately determine the resonant frequency and quality factor of series and parallel resonances.

### 2.5. Determination of the Material Constants Using the Equivalent Circuits

For the theoretical analysis of the frequency dependences of the electrical impedance of the free piezoelectric resonator, the equivalent electromechanical circuit [12] shown in Figure 5 was used.

The equivalent circuit consists of the electromechanical transformer and the electrical and mechanical parts. The electrical part includes the primary winding of the transformer and electrical capacitances *C*_0_ and −*C*_0_. The mechanical part consists of the secondary winding, and the acoustic impedances of the piezoelectric plate *Z*_1_ and *Z*_2_ and electrodes *Z*_1_*^e^* and *Z*_2_*^e^* [12]. The mechanical part of the equivalent circuit corresponding to the air—electrode interface can be considered as shorted due to the low acoustic impedance of the air. The mechanical impedances of the piezoelectric *Z*_1_ and *Z*_2_ and the electrode *Z*_1_*^e^* and *Z*_2_*^e^*in the equivalent circuit are expressed as follows [14]:*Z*_1_ = *izS* tg(*kd*/2)(1)
*Z*_2_ = −*izS*/sin(*kd*/2)(2)
*Z*_1_*^e^* = *iz^e^S* tg(*k^e^d^e^*/2)(3)
*Z*_2_*^e^* = −*iz^e^S*/sin(*k^e^d^e^*/2)(4)

Here, *S* is the area of the electrodes, *k* and *k^e^* are the wave numbers for the materials of the piezoelectric and the electrode, *d* and *d^e^* are the thickness of the piezoelectric plate and the electrodes, *i* is the imaginary unit, and *z* and *z^e^* are the specific mechanical impedances of the materials of the piezoelectric and electrodes, which are expressed as follows:*z* = {(*C* + *e*^2^/*ε* + *iωη*)*ρ*}^1/2^(5)
*z^e^* = {(*C^e^* + *iωη^e^*)*ρ^e^*}^1/2^(6)

Here, *C* is the modulus of elasticity, *e* is the piezoconstant, *ε* is the permittivity, *η* is the viscosity coefficient, *ρ* is the density, and *ω* is the circular frequency. The index “*e*”, as before, means that the quantity belongs to the electrode. Let us also write expressions for the velocity of the acoustic wave in the piezoelectric (*v*) and electrodes (*v^e^*):*υ* = {(*C* + *e*^2^/*ε* + *iωη*)/*ρ*}^1/2^(7)
*υ^e^* = {(*C^e^* + *iωη^e^*)/*ρ^e^*}^1/2^(8)

Assuming that the thickness of the piezoelectric plate is much less than the diameter of the electrode, the electric capacitance can be represented as *C*_0_ = *εS*/*d*. The transformation ratio is *N* = *hC*_0_, where *h* = *e*/*ε* [14]. Further, in the expressions (1)–(8) for the shear wave in the quartz crystal, *C* = *C*_66_, *e* = *e*_16_, *η* = *η*_66_, and *ε* = *ε*_11_ in the rotated coordinate system, corresponding to the AT cut. In this case, the **X**_1_ axis, which is normal to the plate surface, determines the direction of wave propagation, and the **X**_2_ axis is parallel to the wave polarization vector. Accordingly, for the isotropic electrode material, *C^e^* = *C*_66_*^e^* and *η^e^* = *η*_66_*^e^*.

### 2.6. Determination of the Material Constants of the Piezoelectric Material of the Quartz Resonator

First, the modulus of elasticity *C*_66_, the viscosity coefficient *η*_66_, the piezoelectric constant *e*_16_, and the permittivity *ε*_11_ of quartz were determined, corresponding to the chosen coordinate system. For this, the frequency dependences of the real and imaginary parts of the electrical impedance of the free resonator were measured. By analyzing the equivalent circuit (Figure 5) using the Kirchhoff equations and the designations (1)–(8), a program was built that allows for the calculation of the frequency dependences of the real and imaginary parts of the electrical impedance at the input of the electrical part of the circuit. The initial values of the resonator material parameters for the specified coordinate system are presented in Table 1 [17].

Then, using the least squares method, the objective function *F* was constructed using 7 points of the frequency range:(9)$$F=\sqrt{\sum_{n=1}^{7}\left[{\left(R_{theor}^{n}-R_{\exp er}^{n}\right)}^{2}+{\left(X_{theor}^{n}-X_{\exp er}^{n}\right)}^{2}\right]}$$

Here, *R_theor_^n^* and *R_exper_^n^* are the theoretical and experimental values of the real part of the electrical impedance at the *n*th step and *X_theor_^n^* and *X_exper_^n^* are the theoretical and experimental values of the imaginary part of the electrical impedance at the *n*th step.

By the enumeration of the unknown constants *C*_66_, *η*_66_, *e*_16_, and *ε*_11_ of quartz, the set of their parameters corresponding to the minimum of the objective function was determined. In this case, the experimental frequency dependences of the real (*R*) and imaginary (*X*) parts of the electrical impedance of the resonator coincided with the theoretical ones as much as possible (Figure 6). The obtained values of these quantities are presented in Table 2.

Thus, a good agreement was obtained between the theoretical and the experimental dependences for the unloaded resonator, indicating the possibility of using this technique to determine the material constants of a medium that is in reliable acoustic contact with two sides of a resonator [12,14].

### 2.7. Determination of the Material Constants of the Suspensions under Study

At this stage, the determination of the mechanical parameters of the suspensions is based on the use of the equivalent circuit of the resonator loaded on both sides (Figure 7) and the material constants for quartz obtained at the first stage.

For the equivalent circuit shown in Figure 7, the acoustic impedance *Z^s^* and acoustic wave velocities *υ^s^* in the suspension are expressed as:*Z^s^* = *z^s^S*(10)
*υ^s^* = {(*C*_66_*^s^* + *iωη*_66_*^s^*)/*ρ^s^*}^1/2^(11)
*z^s^* = {(*C*_66_*^s^* + *iωη*_66_*^s^*)*ρ*^s^}^1/2^(12)

Here, the belonging of the quantity to the suspension is indicated by the index *^s^*.

By analyzing the equivalent circuit of the loaded resonator (Figure 7) using the Kirchhoff equations and the designations (1)–(8) and (10)–(11), a program was built that allows for the determination of the frequency dependences of the real and imaginary parts of the electrical impedance at the input of the electrical part of the circuit. In analogy with the free resonator, using the least squares method, the objective function *F* (9) was built using 7 points of the frequency range. By the enumeration of the unknown constants *C*_66_*^s^* and *η*_66_*^s^* of the suspension, the pair of the parameters that corresponded to the minimum of the objective function was determined. In this case, the experimental frequency dependences of the real (*R*) and imaginary (*X*) parts of the electrical impedance of the loaded resonator coincided with the theoretical ones as much as possible. Figure 8 demonstrates a good agreement between the theoretical and experimental frequency dependences of the real (*R*) and imaginary (*X*) parts of the electrical impedance of the quartz resonator immersed in pure glycerol.

For all other samples of the suspensions, a good agreement was also observed between the theoretical and experimental dependences of the frequency dependences of the real and imaginary parts of the electrical impedance.

## 3. Results and Discussion

Thus, as a result of the performed work, the values of the shear modulus of the elasticity *C*_66_*^s^*, shear viscosity coefficient *η*_66_*^s^*, and shear acoustic wave velocity *υ^s^* were obtained for the suspension samples with different volume concentrations of synthetic diamond particles. These data are presented in Table 3. To test this approach for determining the viscosity coefficient of the suspensions under study, the shear coefficient of the viscosity of these suspension samples was directly measured by an SV-10 viscometer (A&D Company, Tokyo, Japan). The measured coefficients of the viscosity are indicated as *η*_66_*^s^*^1^ in Table 3.

Thus, it has been shown that the shear modulus of the elasticity (*C*_66_*^s^*) of the studied suspensions first decrease until the volume concentration of particles is 0.098%, then increase with the increase in the volume concentration of the diamond particles (Figure 9). However, this jump is insignificant, and it is comparable with the error of the method. The shear viscosity coefficient *η*_66_*^s^*, determined using the resonator (Figure 10, blue color), first decreases with the increase in the volume concentration of the diamond particles until the value of 0.147%, after which it then monotonically increases. Figure 10 also shows the dependence of the viscosity coefficient obtained by the viscometer (green color).

Figure 10 shows a good agreement between the theoretical dependence (blue color) of the shear viscosity coefficient obtained using the quartz resonator and the experimental dependence (green color) that was obtained by using the viscometer.

We still cannot explain the reason for the existence of a minimum in the dependence of the shear viscosity on the volume concentration of the diamond particles, and this problem requires further research.

The velocity of the shear acoustic wave in the suspension was theoretically determined according to Equation (11) using the obtained values of the density *ρ_s_*, shear modulus *C*_66_*^s^*, and shear viscosity coefficient *η*_66_*^s^* presented in Table 3. Figure 11 shows that the wave velocity decreases with the increase in the particle concentration to the value of 0.147%, where it then increases. The velocity of such a wave is extremely low, and it quickly decays as it propagates.

An analysis of the graphs presented in Figure 3, Figure 9 and Figure 10 clearly shows that the experimentally obtained dependences of the characteristics of the quartz resonator immersed in the suspension (Figure 3) qualitatively correspond to the experimentally obtained curve for the shear viscosity and shear modulus of the elasticity of the suspensions (Figure 9 and Figure 10). Thus, we can conclude that the changes in characteristics of the quartz resonator such as the frequencies and quality factors of the parallel and series resonances and the maximum values of the electrical impedance and admittance as a result of the change in the volume concentration of the particles is qualitatively explained by the change in the shear viscosity coefficient and shear modulus of the elasticity of these suspensions. It should also be noted that, as is known, the shear acoustic waves practically do not propagate in liquids due to the very low shear modulus of elasticity. However, at high viscosities, so-called “viscous” waves can appear in them, which quickly attenuate during the propagation.

## 4. Conclusions

Using an E4990A impedance meter (Keysight Technologies, Santa Rosa, CA, USA) we measured the frequency dependences of the real and imaginary parts of the electrical impedance of the free resonator, as well as of the resonator, which was completely immersed in suspension. We obtained the dependences of the frequency and quality factor of the parallel and series resonances as well as the maximum values of the electrical impedance and admittance on the volume concentration of the particles in the suspension used for the quartz resonator. For the langasite resonator, the similar dependences were only presented for the maximum values of the electrical impedance and admittance; no significant changes were observed for other characteristics. To determine the modulus of elasticity and the viscosity coefficient, we first measured the frequency dependences of the real and imaginary parts of the electrical impedance of the resonator immersed in the suspension. Then, the shear modulus of the elasticity and the shear coefficient of the viscosity of the studied suspensions were determined. This procedure was carried out using Mason’s electromechanical equivalent circuit. The equivalent circuit included the electromechanical transformer and the primary electrical and secondary mechanical circuits. The material constants of the suspensions were determined by fitting the theoretical frequency dependences of the real and imaginary parts of the electrical impedance of the resonator to the experimentally measured ones, which was calculated using the equivalent circuit.

As a result, it was found that the change in the volume concentration of the diamond particles in the suspension leads to the change in its density, shear modulus of the elasticity, shear viscosity coefficient, and velocity of the shear acoustic wave. In addition, during the experiment, the shear viscosities of the obtained suspensions were measured using an SV-10 liquid viscometer (A&D Company, Tokyo, Japan). It was shown that the behaviors of characteristics of the quartz resonator, such as the frequencies and quality factors of the parallel and series resonances and the maximum values of the electrical impedance and admittance of the AT-quartz resonator with the change in the volume concentration of the synthetic diamond particles, qualitatively coincides with changing the shear viscosity coefficient and shear modulus of the elasticity of these suspensions.

## Figures and Tables

**Figure 1 sensors-23-00608-f001:**
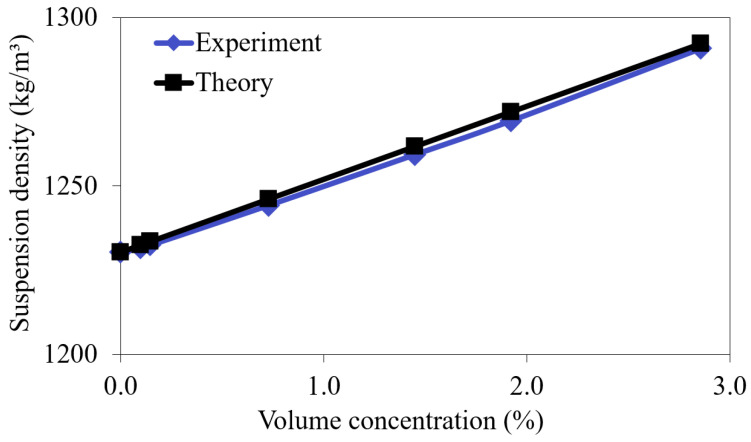
The theoretical and experimental dependences of the suspension density on the volume concentration of the diamond particles.

**Figure 2 sensors-23-00608-f002:**
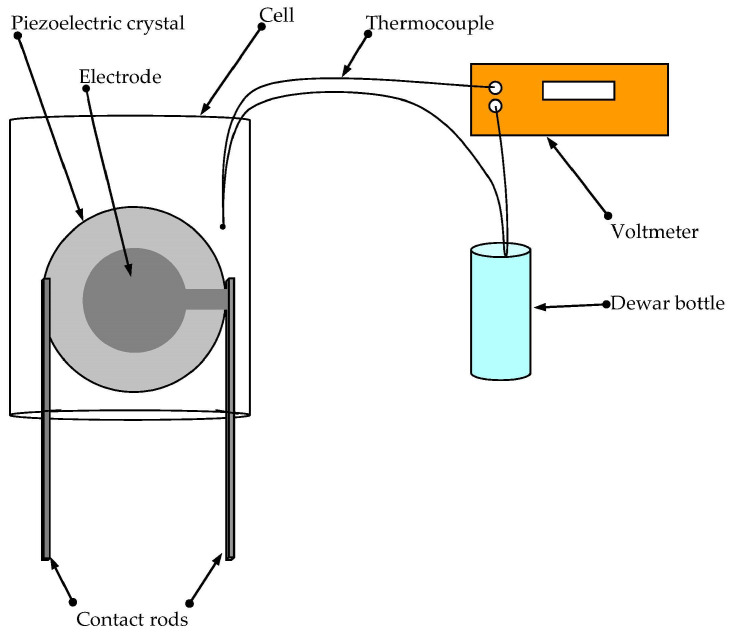
The experimental setup for carrying out the research with the suspensions.

**Figure 3 sensors-23-00608-f003:**
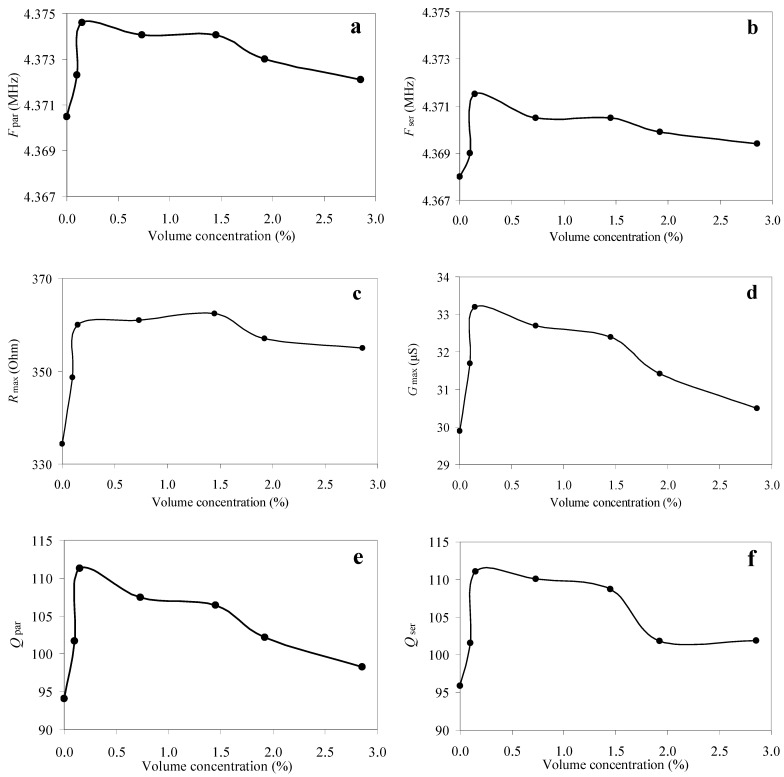
The dependences of the resonant frequency of the parallel (**a**) and series (**b**) resonances, the maximum values of the real parts of the electrical impedance (**c**) and admittance (**d**), as well as the quality factor of the parallel (**e**) and series (**f**) resonances of the quartz resonator on the volume concentration of the diamond particles in the suspension.

**Figure 4 sensors-23-00608-f004:**
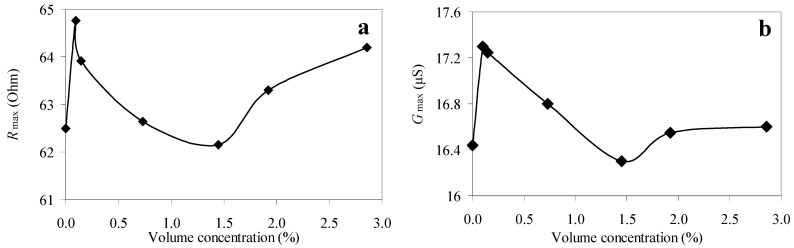
The dependences of the maximum values of the real parts of the electrical impedance (**a**) and admittance (**b**) of the langasite resonator on the volume concentration of the diamond particles in the suspension.

**Figure 5 sensors-23-00608-f005:**
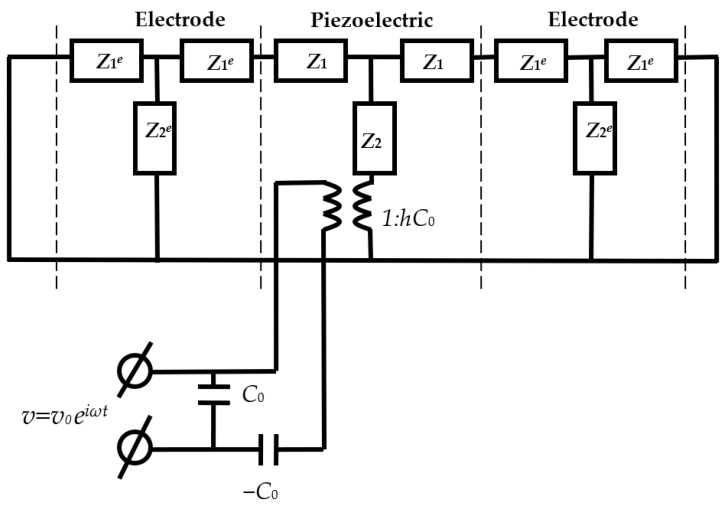
Equivalent circuit of the resonator with the electrodes without load.

**Figure 6 sensors-23-00608-f006:**
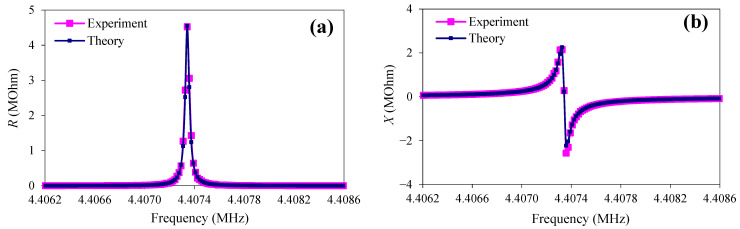
The frequency dependences of the real (**a**) and imaginary (**b**) parts of the electrical impedance of the AT-quartz resonator without load (pink—experiment; blue—theory).

**Figure 7 sensors-23-00608-f007:**
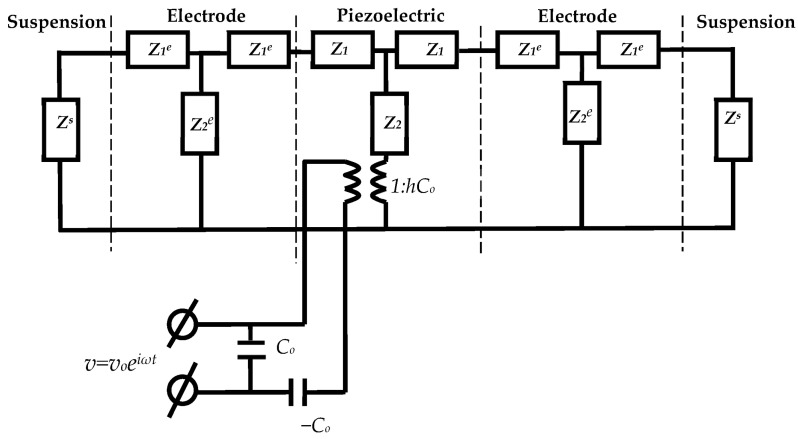
Equivalent circuit of the resonator with the electrodes loaded on both sides.

**Figure 8 sensors-23-00608-f008:**
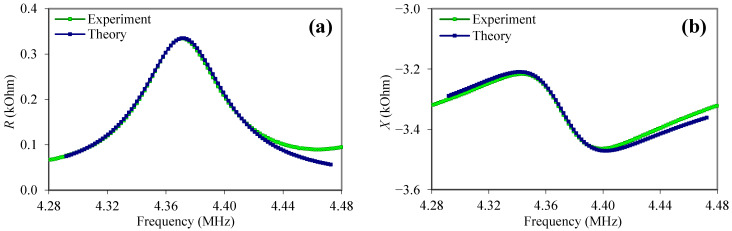
Frequency dependences of the real (**a**) and imaginary (**b**) parts of the electrical impedance of the quartz resonator loaded on both sides with glycerol (blue color—theory; green color—experiment).

**Figure 9 sensors-23-00608-f009:**
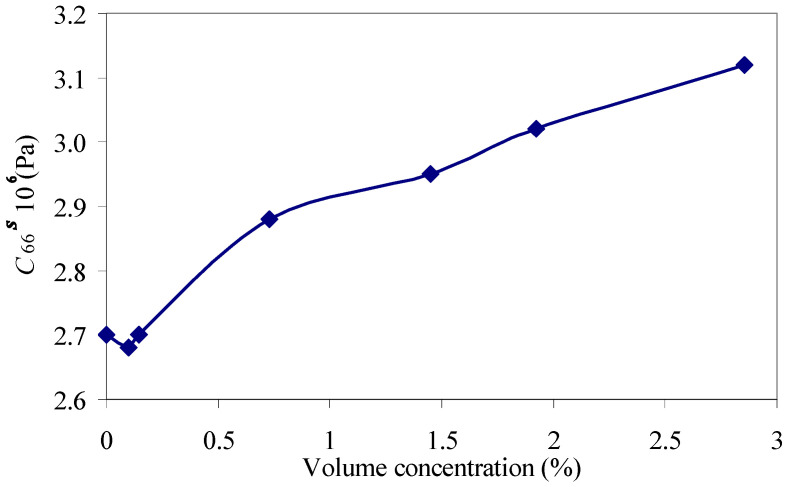
The dependence of the shear modulus of the elasticity C66s of the suspension on the volume concentration of diamond particles with particle sizes of 1–2 µm.

**Figure 10 sensors-23-00608-f010:**
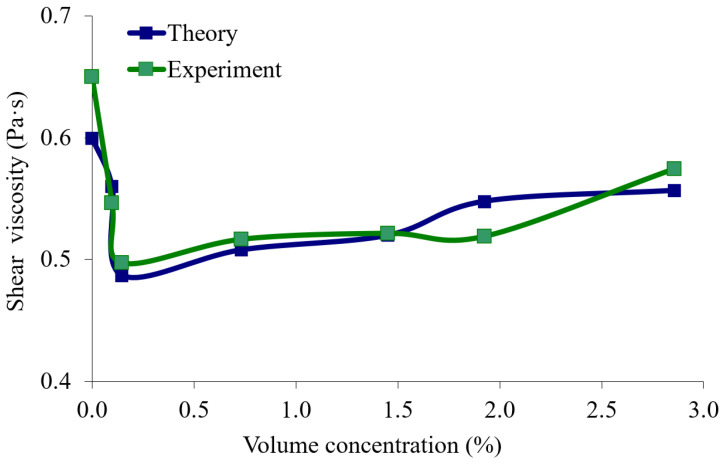
The dependence of the shear viscosity coefficient of the suspension on the volume concentration of the diamond particles. The data obtained by the resonator (blue color) and by the viscometer (green color).

**Figure 11 sensors-23-00608-f011:**
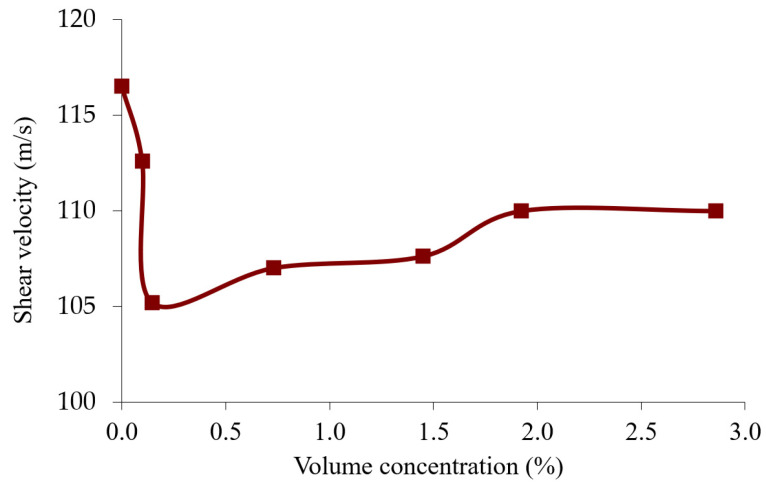
Dependence of the velocity of the shear acoustic wave in the suspension on the volume concentration of the diamond particles (according to the data in Table 3).

**Table 1 sensors-23-00608-t001:** The initial values of the quartz resonator material constants.

*ρ*, kg/m^3^	*C*_66_, 10^10^ Pa	*η*_66_, Pa·s	*e*_16_, C/m^2^	*ε*_11_, 10^−11^ F/m
2648.38	2.899	0	0.067	3.92

**Table 2 sensors-23-00608-t002:** The obtained values of the quartz resonator material constants.

*ρ*, kg/m^3^	*C*_66_, 10^10^ Pa	*η*_66_, Pa·s	*e*_16_, C/m^2^	*ε*_11_, 10^−11^ F/m
2649	2.892	0.009	0.0671	4.05

**Table 3 sensors-23-00608-t003:** The obtained values of the material constants and shear acoustic wave velocity for the suspension samples at different volume concentrations of synthetic diamond particles.

Volume Concentration, %	*ρ^s^*, kg/m^3^	*C*_66_*^s^*, 10^6^ Pa	*η*_66_*^s^*, Pa·s	*υ^s^*, m/s	*η*_66_*^s^*^1^, Pa·s
0.0	1230.3	2.7	0.6	116.5	0.65
0.098	1231.6	2.68	0.56	112.6	0.546
0.147	1232.44	2.7	0.487	105.2	0.497
0.73	1244.27	2.88	0.508	107	0.517
1.45	1259.3	2.95	0.52	107.6	0.522
1.923	1269.35	3.02	0.548	110	0.519
2.857	1290.8	3.12	0.557	110	0.575

## Data Availability

The authors confirm that this research is new and has not been published anywhere.

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
