# Peer review of "The Effect of Glycerol-Based Suspensions on the Characteristics of Resonators Excited by a Longitudinal Electric Field"

_sensors, 2023, doi:10.3390/s23020608_

Round 1
Reviewer 1 Report
The paper “The effect of suspensions based on glycerol on the characteristics of resonators excited by longitudinal electric field” uses bulk acoustic waves generated in AT-cut quartz crystal to measure mechanical(elastic) properties of suspensions of diamond powder in glycerol. The volume concentration of diamond microparticles is up to 3%.
With their straightforward experimental procedure the authors are able to assess the elastic properties of diamond suspension like shear modulus and velocity, density and shear viscosity coefficient.
Before my recommendation for publication, the authors must respond to the following issues:
-1) the concentration of the suspended diamond is quite small (max. 3%). In that case, all calculated elastic properties should vary linearly with powder concentration. Experimental data show, however, a nonlinear behavior. Apparently the linear trend is only up to 0.147%, which is quite low. How can be explained by any theory of mixing properties in this suspension, thinking of it as a composite?
-2) Can the authors explain why there is no sensibility for langasite? Eventually by measuring the electro-mechanical parameters of langasite and compare them with the quartz parameters?
-3) Due to the fact that there is no monotonic relationship between concentration and elastic properties of the suspension, how the experimental setup can be used to measure the concentration of diamond microparticles?
There is also a minor issue: Ref. 9 and 17 are duplicate.
Author Response
-1) the concentration of the suspended diamond is quite small (max. 3%). In that case, all calculated elastic properties should vary linearly with powder concentration. Experimental data show, however, a nonlinear behavior. Apparently the linear trend is only up to 0.147%, which is quite low. How can be explained by any theory of mixing properties in this suspension, thinking of it as a composite?
Thank you for the interesting note.
As for the elastic modulus, it increases monotonically with an increase in the volume concentration of the diamond particles, as shown in Fig. 9. An exception is the starting point corresponding to the pure glycerol. However, this jump is insignificant and it is comparable with the error of the method.
Regarding the shear viscosity. Firstly, the viscosity was determined by fitting the theoretical frequency dependencies of the real and imaginary parts of the electrical impedance resonator, immersed in the suspension, to the experimental ones. This method showed the presence of a pronounced minimum on the dependence of viscosity on the volume concentration of the diamond particles (Fig. 10). The shear viscosity of the suspensions was also measured using the SV-10 viscometer, and this method also showed the presence of the specified minimum. It should be noted that at first the authors forgot to take into account that the values of the viscosity of the suspension samples measured using the SV-10 viscometer must be divided into the density of the sample, as indicated in the guidance to the viscometer. After correcting this error, the dependences of the viscosity on the volume concentration of the diamond particles, measured by two methods, turned out to be the same. In the new version of the article, these dependencies are shown in the corrected figure 10. Thus, both methods have shown the presence of a minimum at a concentration of 0.147%.
We have included the following text after Figure 10:
“Figure 10 shows a good agreement between the theoretical dependence (blue color) of the shear viscosity coefficient obtained using the quartz resonator and experimental dependence (green color) that was obtained by using viscometer.”
As for the minimum on the dependence of the shear viscosity on the volume concentration of diamond particles. We find it difficult to explain the existence of a minimum in the dependence of the shear viscosity on the volume concentration of the diamond particles. We tried to draw a parallel with the results of the well-known work [5], in which the dependence of the attenuation coefficient of a longitudinal acoustic wave on the viscosity of a liquid with the solid particles was calculated. They have shown that this dependence has a minimum depending on the particle size and concentration. However, in our case, we have a minimum of shear viscosity depending on the volume concentration of particles. In addition, in [5], a longitudinal acoustic wave is analyzed, and we consider the shear oscillation of the resonator surface, which is in contact with the suspension under study. Therefore, we have added the following sentence on the page 11.
“We still cannot explain the reason for the existence of a minimum in the dependence of the shear viscosity on the volume concentration of the diamond particles, and this problem requires further research.”
-2) Can the authors explain why there is no sensibility for langasite? Eventually by measuring the electro-mechanical parameters of langasite and compare them with the quartz parameters?
Thank you for this comment
The resonant characteristics of the quartz and langasite resonators differ greatly for the following reason. A quartz resonator resonates on a shear acoustic wave. Therefore, the contact of the resonator with the suspension does not lead to the emission of an acoustic wave into the suspension due to the absence of a mechanical displacement component normal to the surface. Thus, due to the absence of the radiation losses, the resonator retains a sufficiently high quality factor in the contact with the suspension, which allows to accurately determine the resonant frequency and the quality factor of the series and parallel resonances. The langasite resonator resonates on a longitudinal acoustic wave. Therefore, the contact of the resonator with the suspension leads to the emission of a longitudinal acoustic wave into the suspension due to the presence of a mechanical displacement component normal to the surface. Thus, radiation losses during the contact of the resonator with the suspension significantly reduce the quality factor up to 16, which does not allow one to accurately determine the resonant frequency and the quality factor of the series and parallel resonances. Estimates showed that the relative change in the frequency of the parallel and series resonances with a change in the concentration of particles in the range of 0.1-2.86% did not exceed 0.015%. In this case, no regularity in the change in the frequency and quality factor with a change in the concentration of the particles was observed. We have included the following paragraph in the text in the paper on page 5:
“This is due to the fact that, unlike a quartz resonator, a langasite resonator resonates on a longitudinal acoustic wave. Therefore, the contact of the resonator with the suspension leads to the emission of a longitudinal acoustic wave into the suspension due to the presence of a mechanical displacement component normal to the surface. Thus, the radiation loss during the contact of the resonator with the suspension significantly reduces the quality factor up to 16, which does not allow one to accurately determine the resonant frequency and the quality factor of series and parallel resonances.”
-3) Due to the fact that there is no monotonic relationship between concentration and elastic properties of the suspension, how the experimental setup can be used to measure the concentration of diamond microparticles?
The experimental setup presented in this work was not planned to measure the unknown concentration of diamond microparticles in the suspension. The purpose of the work is to determine the patterns of change in the viscosity and elastic modulus of a suspension based on glycerol with a change in the volume concentration of diamond particles.
4) There is also a minor issue: Ref. 9 and 17 are duplicate.
We have corrected this mistake.

Reviewer 2 Report
Please refer attached file for comments.

Author Response
Line 305 to 308
To test this approach for determining the elastic modulus and viscosity coefficient of the suspensions under study, the shear coefficient of the viscosity of these suspension samples was measured directly by viscometer SV – 10 (A&D Company, Tokyo, Japan).
Thank you for this note. We have corrected this sentence:
To test this approach for determining the elastic modulus and viscosity coefficient of the suspensions under study, the shear coefficient of the viscosity of these suspension samples was measured directly by viscometer SV – 10 (A&D Company, Tokyo, Japan).
Line 310 to 311
Table 3. The obtained values of material constants and shear acoustic wave velocity for suspension samples at different volume concentrations of synthetic diamond particles.
We have corrected this sentence:
Table 3. The obtained values of material constants and shear acoustic wave velocity for suspension samples at different volume concentrations of synthetic diamond particles.
Line 328
According to the authors, the difference…
“authors” mentioned in this sentence, are they referred to previous authors? please cite the reference.
If “authors” mentioned are referring to the authors of this article, the sentence should be rephrased.
We have removed this sentence because we have corrected an error when measuring shear viscosity with a viscometer. It should be said that at first the authors forgot to take into account that the values of the viscosity of the suspension samples measured using the SV-10 viscometer must be divided into the density of the sample, as indicated in the guidance to the viscometer. After correcting this error, the dependences of the viscosity on the volume concentration of the diamond particles, measured by two methods, turned out to be the same. In the new version of the article, these dependencies are shown in the corrected figure 10.
Line 333
There is no discussion related to Figure 11 in the section. Please elaborate the findings in Figure 11.
We have included the following sentence below the Fig. 11:
“The velocity of the shear acoustic wave in the suspension was theoretically determined according to the equation (11) using the obtained values of density ρs, shear modulus C66s and shear viscosity coefficient η66s, presented in Table 3. Figure 11 shows, that the wave velocity decreases with increasing the particle concentration to the value 0.147% and then increases. The velocity of such a wave is extremely low, and it quickly decays as it propagates.”

Reviewer 3 Report
The paper investigates the effect of pure glycerol-based suspensions and diamond powders of different concentrations on the characteristics of resonators under longitudinal electric fields, and the results are of interest for the development of resonator applications. The drawback of the paper is that the abstract and conclusions are not empirical enough and the authors are advised to condense them.
Author Response
The paper investigates the effect of pure glycerol-based suspensions and diamond powders of different concentrations on the characteristics of resonators under longitudinal electric fields, and the results are of interest for the development of resonator applications. The drawback of the paper is that the abstract and conclusions are not empirical enough and the authors are advised to condense them.
Thank you for this comment
These sections have been shortened.

Round 2
Reviewer 1 Report
The authors improved the manuscript, now the paper can be published.